# Umbilical cord separation time, predictors and healing complications in newborns with dry care

**María Dolores López-Medina**[1,2]☯*, **Ana Belén López-Araque**[3]☯, **Manuel Linares-Abad**[1]‡, **Isabel María López-Medina**[1]‡

**1** Department of Nursing, Faculty of Health Sciences, Universidad de Jaén, Jaén, Spain, **2** San José Health Center, Northern Jaén Sanitary District, Linares, Jaén, Spain, **3** Hospital of Jaén, Andalusian Health Service, Jaén, Spain

☯ These authors contributed equally to this work.
‡ These authors also contributed equally to this work.
* mlmedina@ujaen.es

**Data Availability Statement:** All relevant data are within the manuscript and its Supporting Information files.

## Abstract

### Objective

The objective of this study was to explore the umbilical cord separation time, predictors, and healing complications from birth until the newborn was one month old.

### Design

A quantitative longitudinal observational analytical study by stratified random sample was adopted.

### Setting

Public health system hospitals in southern Spain and at newborns' homes.

### Participants

Between April 2016 and December 2017, the study included 106 neonates born after 35–42 weeks of gestation whose umbilical cord was cured with water and soap and dried later as well as newborns without umbilical canalisation whose mothers enjoyed a low-risk pregnancy.

### Methods

The data collection procedure comprised two blocks: from birth to the time of separation of the umbilical cord and from cord separation to the first month of life of the newborn. Umbilical cord separation time was measured in minutes; socio-demographic and clinical characteristics were measured by means of questionnaires, and the external diameter of the umbilical cord was measured using an electronic stainless-steel calliper and trailing roller.

**Funding:** The author(s) received no specific funding for this work.

**Competing interests:** The authors have declared that no competing interests exist.

## Results

The mean umbilical separation time: 6.61 days (±2.33, IC 95%:6.16–7.05). Incidence of omphalitis was 3.7%; granuloma was 8.6%. Separation time predictors were wetting recurrence, birth weight, intrapartum antibiotics, birth season, and Apgar < 9 ($R^2$ = 0.439 F: 15.361, p <0.01).

## Conclusion

The findings support the World Health Organization recommendations: dry umbilical cord cares is a safe practice that soon detaches the umbilical cord, taking into account the factors studied that will vary the length of time until the umbilical cord is separated.

## Introduction

The umbilical cord, which connects the baby and placenta in the uterus, is made of blood vessels and connective tissue. It is covered by a membrane that is normally bathed in amniotic fluid. After birth, cutting the cord physically and symbolically separates the mother and her baby. The cord stump dries and falls off, and the wound heals [1]. The World Health Organization [2] estimates that a quarter of the world's neonatal deaths are due to infection; 75% of these occur in the first week of life, with the umbilical cord being the gateway [3]. The umbilical cord is an important bacterial colonisation site. A possible consequence of bacterial colonisation is cord stump infection, a factor that can greatly increase morbidity and mortality. The risk of infection increases until the stump detaches [4]. The newborn's skin and umbilical cord begin to colonise with saprophytic bacteria provided by the mother through skin contact [4]. The umbilical cord is colonised by microorganisms of the vagina, skin, and hands of the birth attendant. Umbilical cord rot becomes an excellent growing medium for microbial flora [5]. Omphalitis is an infection of the umbilical cord stump. Tracking bacteria along the umbilical vessels may lead to septicaemia, which can result in neonatal morbidity and mortality [6]. The average age of presentation of omphalitis is the third or fourth day of life of the newborn [5,7–9]. Omphalitis can be extremely serious, causing sepsis, due to the permeability of the umbilical vessels that persists until approximately 20 days of age of the newborn [8]. Prevention of this disease and its associated neonatal mortality is of great public health importance [10].

The exact incidence of cord infections is unknown. They appear to be relatively rare in developed countries, but they are probably under-reported [11]. Overall, omphalitis risk varies substantially and depends on the level of direct and indirect exposures to the stump (e.g., absence of hand washing and other hygiene practices by carers), variation in definition, standardisation, and frequency of measurement [12]. Estimated that the incidence of omphalitis in newborns in developed countries is around 0.7% and rises to 6–8% in developing countries [9,10,13,14]. Even with the low incidence of omphalitis, the open wound of the umbilical stump remains an entry point for pathogenic bacteria, and the undetached cord stump often worries mothers. Rapid healing of the cord is an important aspect of infection prevention [15]. The dry care method ('keep clean and dry') using only soap and water for umbilical cord care has been recommended by the WHO [11]. This recommendation is still in force today.

According to the National Institute of Health Care and Excellence guidelines, parents should be advised how to keep the umbilical cord clean and dry and that antiseptics should not be used routinely [16]. Dry care may be as effective and practical as antiseptics.

Furthermore, dry care would be less expensive for healthy newborns in hospital settings in high-income countries. In the same vein are the recommendations of the American Academy of Paediatrics [17] and the Spanish Association of Paediatrics and Standards Commission of the Spanish Society of Neonatology. These organizations also claim that only water, soap, and drying afterward are necessary for umbilical cord care; the application of antiseptic solutions can delay the separation of the umbilical cord and does not provide any benefit [18]. In high-income countries, where mortality is very low, important outcomes should include infections that occur in the first month of life, maternal satisfaction, and the time it takes for the cord to separate. There is good reason to conduct a study on the least expensive interventions available without specific topical umbilical cord care. Much of the concern of mothers and health professionals relates to uncertainty about the normal process of drying and detachment of the umbilical cord, including its appearance and the smell of decaying tissue [1]. Lack of knowledge leads families and professionals to use antiseptics for umbilical cord care frequently.

The main objective of this research is to analyse the natural separation process of the umbilical cord with dry care and its associated factors in order to determine the mean separation time of the umbilical cord of newborns, to know the incidence of omphalitis and other associated complications in the newborns' first month of life, and to determine what factors predict cord separation time.

## Methods

### Study design

A quantitative longitudinal observational analytical study was conducted to explore the umbilical cord separation time, predictors, and healing complications from birth until the newborn was one month old.

### Setting and sampling

The participants were newborns in the public health system of the province of Jaén in southern Spain. In public hospitals of the province of Jaén, there were a total of 4,739 births in 2015 [19], distributed as follows: south area, 1,979 (41.76%); western area, 514 (10.85%); eastern area, 1,216 (25.66%); and northern area, 1,030 (21.73%). The number of births from high-risk obstetric pregnancies in the same year in the province of Jaén was 904, so the reference population is 3,835.

Inclusion criteria: Newborns who were between 35 and 42 weeks of gestation; who had their umbilical cord cured with water and soap and dried later; and whose mothers were over the age of 18, enjoyed a low-risk pregnancy, and consented to participate in the study.

Exclusion criteria: Newborns with umbilical canalization and language barrier with the mother of the newborn.

A stratified random sample was adopted. To identify the population and select the unit of analysis, a list of pregnant women attending the ultrasound consultation at 33 weeks of gestation was used. Pregnant women were selected randomly and equitably on a monthly basis. After the literature review, we established an error of no more than seven hours in the estimate of a mean separation time and assumed a standard deviation of 37.42 hours; for a population size of 2,619 with correction for finite populations (95% confidence interval, bilateral) a total of 105 neonates would need to be included in the study. First, a probabilistic selection was carried out by clusters of the sample, where the hospitals were selected in the first stage. In the second stage, we selected the mother–newborn binomial to be measured, and a proportional sample stratification was carried out through fh: n/N = ksh [20]. The eastern-area hospital did

not meet the inclusion criteria, because it cares for the cord with alcohol solution, so it was eliminated from the study.

After selecting the participants, researchers informed them about the study through a personal pre-birth visit. Participants signed an informed consent document, and researchers provided a 'reminder sticker' so they would not forget to tell them the day they went into labour.

### Measures

**Dependent variables.** <u>Umbilical cord separation time.</u> Is time when the umbilical cord naturally separates from the infant; measured in minutes from the time of birth to the time of complete separation of the newborn's umbilical cord, by the lead researcher (midwife).

<u>Healing complications</u>. Occurrence of neonatal omphalitis, measured according to the National Healthcare Safety Network belonging to the Centers for Disease Control and Prevention surveillance definitions and criteria [21]; other complications: granuloma, cutaneous navel, umbilical cord hernia, late separation of the umbilical cord, supraumbilical hernia, persistence of the omphalomesenteric duct, congenital anomalies of the urachus, and slight bleeding after separation [8] (measured by midwife observation and derivation for medical diagnosis).

**Independent variables.** <u>Type of care</u>. The different actions carried out in the dry umbilical cord care: how often the umbilical cord gets wet, pH of soap used, type of bath for neonates, wearing a garment that tightens the newborn's body, placing the newborn's diaper under the umbilical cord, and covering the umbilical cord with clean gauze.

<u>External diameter of the umbilical cord</u>. Measured by means of an electronic stainless-steel calliper and trailing roller, with a measuring range up to 150 mm, with an accuracy of 0.02, a measuring mouth of 40 L in mm, a resolution of 0.01 mm/0.0005 in, an operating temperature of 0–40˚C, and a maximum humidity of 80%, with IP54 protection against dust and water. Measurements were performed in the first 12 hours of life of the newborn.

In Table 1 we can see the independent secondary variables of the studied population.

**Data collection procedure.** Data collection was conducted from April 2016 to December 2017. The process was comprised of two blocks:

<u>From birth to the time of separation of the umbilical cord</u>. There were three visits to the newborn: First visit: at the hospital, within the first 12 hours of the newborn's birth; the lead researcher (midwife) collected clinical data from the delivery process and the newborn and performed a clinical evaluation of the umbilical cord and measurement of its external diameter. Second visit: at the family home, three days after birth; the same researcher monitored the umbilical cord clinically with special consideration of the evaluation criteria of omphalitis and care for the umbilical cord. Third visit: at the family home, two days after umbilical cord separation; once again, the same researcher monitored the umbilical cord clinically, paid attention to the occurrence of complications in the healing process, and collected data about how many times the cord got wet and the umbilical cord separation time.

<u>From cord separation to the first month of life of the newborn</u>. Monitoring of the evolution of the healing process. The mothers were previously instructed in writing about the warning signs and normality. The researcher contacted the mothers of the newborns weekly by telephone for information and photos of the umbilicus of the newborns.

**Statistical analysis.** A univariate descriptive analysis of frequencies and percentages was performed for qualitative variables such as the occurrence of complications in healing, and the baseline characteristics of participants, including socio-demographic and clinical information, were summarised (see S1 File). The mean, standard deviation, and median were used to determine the cord separation time and other quantitative variables.

**Table 1. Socio-demographic maternal variables, socio-demographic and clinical newborn variables and labour variables.**

| Variables | Mean ± SD | Mo | N (%) |
|---|---|---|---|
| *Socio-demographic maternal variables* | | | |
| Age (years) | 31.39 ±4.5 | 32 | |
| 18–25 | | | 8(7.5%) |
| 26–35 | | | 81(76.4%) |
| >35 | | | 17(16%) |
| Marital status: | | | |
| Married | | | 76 (72.4%) |
| Single | | | 21 (20%) |
| Divorcee | | | 3 (2.9%) |
| Unmarried partner | | | 5 (4.8%) |
| Education level | | | |
| Higher education | | | 50 (47.6%) |
| High school | | | 32 (30.5%) |
| Job training | | | 18 (17.1%) |
| Primary education | | | 5 (4.8%) |
| Economic status: above the minimum wage | | | 95 (89.6%) |
| Maternal occupational state: Unemployed | | | 20 (19%) |
| Dwelling place: urban dweller | | | 71 (67%) |
| Health departments | | | |
| Jaén | | | 46(43.4%) |
| North Jaén | | | 51 (48.1%) |
| Southern Jaén | | | 6 (5.7%) |
| Northeast Jaén | | | 3 (2.8%) |
| Nationality | | | |
| Spain | | | 102 (96.2%) |
| Colombia | | | 2 (1.9%) |
| Equator | | | 1 (0.9%) |
| United States | | | 1 (0.9%) |
| Chronic pathologies | | | 24 (22.9%) |
| Immunization TT | | | 106 (100%) |
| Smokers | | | 11 (10.5%) |
| Primiparous | | | 59 (55.7%) |
| *Socio-demographic and clinical newborn variables.* | | | |
| Biological sex | | | |
| Female | | | 53 (50%) |
| Male | | | 53 (50%) |
| Newborn weight | 3.342.55gr. (±464.30) | | |
| APGAR 7–8 | | | 19 (17.9%) |
| Feeding | | | |
| Breastfeeding | | | 58 (55.2%) |
| Mixed lactation | | | 43 (41%) |
| Artificial lactation | | | 4 (3.8%) |
| Types of birth | | | |
| Natural birth | | | 69 (65.1%) |
| Instrumental delivery | | | 18 (17%) |
| Cesarean | | | 19 (17.9%) |
| Gestational age | 40 weeks (±7,95 days) | | |

(*Continued*)

**Table 1.** (Continued)

| Variables | Mean ± SD | Mo | N (%) |
|---|---|---|---|
| **Birth pathology** | | | |
| **Without pathology** | | | 92 (86.8%) |
| **Jaundice** | | | 6 (5.7%) |
| **Hypoglycaemia** | | | 4 (3.8%) |
| **Kidney atrophy** | | | 1 (0.9%) |
| **Metabolopathy** | | | 1 (0.9%) |
| **Immaturity** | | | 1 (0.9%) |
| **Inguinal abscess** | | | 1 (0.9%) |
| Season of the year | | | |
| **Spring** | | | 36 (34%) |
| **Summer** | | | 26 (24.5%) |
| **Fall** | | | 26 (24.5%) |
| **Winter** | | | 18 (17%) |
| **Cord diameter** | 9.85mm (5.13–19.10) | | |
| *Labour variables* | | | |
| **Streptococcus agalactiae positive** | | | 14 (13.2%) |
| **Sexually transmitted infection** | | | 0 (0%) |
| **Premature rupture of membranes >18 hours.** | | | 16 (15.1%) |
| **Fever > 38˚C** | | | 7 (6.6%) |
| **Intrapartum antibiotics** | | | 44 (41.5%) |
| **Chorioamnionitis** | | | 0 (0%) |
| **Skin-to-skin contact at birth** | | | 10 (9.4%) |
| **Length of hospital stay** | 54hours (±27.42) | | |
| **Time of the second stage of labour** | 60 minutes (± 60) | | |

Note. SD, standard deviation; Mo, mode; TT, tetanus toxoid; mm, millimetres; ˚C, degrees Celsius.

A bivariate analysis was performed to compare dependent and independent variables and dependent endpoints with each other. Diagnosis of normality was previously made through the Kolmogorov–Smirnov test to determine the use of parametric or nonparametric tests. To determine the association between different umbilical cord care and cord separation time and to determine the association between umbilical cord separation time and socio-demographic and clinical variables, the Student's t, Mann–Whitney-U, and Rho-Spearman tests were used, depending on the hypothesis contrast. To contrast the association between the cord separation time and cord diameter, the R-Pearson test was used. To measure the association among different umbilical cord care, socio-demographic and clinical variables, and healing complications, depending on the hypothesis contrast, the following were used: Fisher exact test, simple binary logistic regression, and Chi Square test.

A multivariate analysis using multiple linear regression was carried out to determine the predictors of umbilical cord separation time. The independent variables were operationalised, according to requirements of coding the independent variables in multiple regression models, and the polytomous independent variables were established as dummy variables. The continuous independent variables were maintained with the original numerical value, and the dichotomous variables was coded as 1 and 0. The backward stepwise regression method was used in this study. We included in the analysis those variables that correlated in the bivariate analysis and the variables recommended by the literature. The precaution was not to include an

excessive number of variables according to sample size [22]. Diagnostic testing of the model was performed, and the following criteria were met: independence of residues: Durbin–Watson; residue normality: Kolmogorov–Smirnov residue; linearity: seen with partial regression graphs; homoscedasticity: graph of standardised residue; non-collinearity: variance inflation factor and tolerance. Analysis of the extreme values was performed with Cook's Distance.

All statistical analyses were conducted using IBM SPSS version 24 [23]. P values of .05 were considered statistically significant.

### Ethics

The study was conducted in accordance with the basic principles for all medical research [24], respecting the applicable legal precepts regarding the protection of personal data, as well as the rights and obligations referred to health information, healthcare records, and documentation. Jaén Research Ethics Committee (26 November 2015) approved the study. Considerations such as confidentiality, voluntary participation, and full information on the nature of the study were extended to all participants, and we obtained the informed consent of the women to participate. The women were reassured that their participation—or lack thereof—would have no influence on their clinical care.

### Limitations

The conditions to which the umbilical cord was exposed could not be controlled, such as atmospheric conditions (temperature, relative humidity, pressure, wind, and rainfall), the level of hygiene of those who were in contact with the neonates, or the level of asepsis when cutting the umbilical cord at birth by the delivery professional.

## Results

Difficulties were encountered in capturing the sample in the western zone, so we reluctantly decided to reject this population group. The sample size for the new population was recalculated, giving a total of 106 newborns, 55 newborns from the southern area and 51 newborns from the northern area.

Table 1 shows socio-demographic maternal variables, and socio-demographic clinical newborn variables and labour variables.

### Umbilical cord separation time

The mean umbilical separation time was 6.61 days (±2.33, IC 95%:6.16–7.05), and the median was 6.14 days. Fifty percent ($P_{25}$-$P_{75}$) of the cords were separated between 4.97 and 7.65 days. The lowest and the highest umbilicus separation times were 1.97 and 13.96 days, respectively (Table 2).

When the cord separation time was compared with the different umbilical cord care, the findings demonstrated a statistically significant association with a large correlation force between separation time and how often the umbilical cord gets wet, with a statistical power of

**Table 2. Umbilical cord separation time.**

| Cord separation time | Mean (SD) | Median | Confidence interval |
|---|---|---|---|
| **Total newborns** | 6.61 days (±2.33) | 6.14 | 6.16–7.05 |
| **Newborns living in north** | 6.64 days (±2.44) | 6.41 | 6.07–7.21 |
| **Newborns living in the south** | 6.53 days (±2.08) | 6.05 | 5.79–7.26 |

100%. Dividing the sample into those umbilical cords that got wet 0–4 times compared to those that got wet $\geq$ 5 times, the findings were as follows: Student's t = -4.102, p value = 0.000. The mean of the group that wet the umbilical cord 0–4 times (8,548.68 min) was lower than the mean of the group that wet the cord $\geq$ 5 times (11,116.20 min), which is a statistically significant result with a statistical power of 97% (Cohen's d = 0.83). There was a large effect size and negative association; the r = 0.43 and $r^2$ = 0.18, so 79% of the 0–4 group was below the mean of the $\geq$ 5 group. The probability that a neonate in the $\geq$ 5 group will have a higher mean separation than another neonate in the 0–4 group if both are chosen at random is 47.4%. In Table 3 we can see the correlation between umbilical cord care and separation time.

The outcomes were the following in the association between umbilical cord separation time and socio-demographic and clinical variables of newborns (Table 4)

Some variables bordered on statistical significance: breastfeeding (p value = 0.07, t: -1.830), birth weight (p value = 0.055, R: 0.187), Apgar (p value = 0.06, R: -0.182), and season of birth: spring (p value = 0.06, t: 1.898).

There was no statistically significant association between separation time and other variables, such as cord diameter (p value = 0.865, R: 0.017), premature rupture of membranes >18 hours (p value = 0.467, Z: -0.728), fever > 38˚C (p value = 0.337, Z: -0.960), skin-to-skin contact at birth (p value = 0.524, Z: -0.638), biological sex (p value = 0.121. Z: -1.551), gestational age (p value = 0.890, R: -0.14), birth pathology (p value = 0.922, Z = -0.097), education level (p value = 0.122, t: 1.56), economic status (p value = 0.816, Z: -0.233), health departments: north/south (p value = 0.826, t: -0.220), birth hospital (p value = 0.615, t: -0.505), or ethnicity (p value = 0.726, Z: -0.350).

No statistically significant differences were found between umbilical cord separation time and the occurrence of complications, such as slight bleeding (p value: 0.154, t: -1.435), umbilical granuloma (p value = 0.436, U: 364), umbilical hernia (p value = 0.710, U: 135), omphalitis (p value = 0.226, U: 131), and mild umbilical redness (p value = 0.908, t: 0.115).

## Umbilical cord healing complications

Regarding the appearance of complications in the healing of the umbilical cord, the incidence of omphalitis was mild omphalitis at 2.8% (three cases, who needed antiseptics or antibiotic creams but did not need hospitalisation; these cases of omphalitis were diagnosed because of the clinical symptomatology that the neonates presented) and moderate omphalitis at 0.9% (one case, who required hospitalisation and treatment with antiseptics and intravenous and local antibiotics).

**Table 3. Umbilical cord care and separation time.**

| Contrast of variables with separation time | Indicators of effect | p-value (statistical power) |
|---|---|---|
| **How often the umbilical cord gets wet [a]** | 0.591 | **0.000 (100%)** |
| **Cover umbilical cord with clean gauze [b]** | 0.743 | 0.459 (12%) |
| **Cover the umbilical cord with diaper [c]** | -0.493 | 0.622 (21%) |
| **Umbilical cord uncovered [c]** | -1.77 | 0.077 (38%) |
| **Wearing a tight-fitting T-shirt** | -1.021 | 0.307 (11%) |
| **Underwater bath [c]** | -0.714 | 0.475 (3%) |
| **Shower bath [b]** | -0.647 | 0.519 (10%) |

[a]Rho-Spearman test,
[b]Student's t,
[c]Mann–Whitney-U.

**Table 4. Socio-demographic and clinical variables of newborns and separation time.**

| Contrast of variables with separation time | Indicators of effect | Effect size | Coefficient of determination $R^2$ | p-value |
|---|---|---|---|---|
| Vaginal delivery [a] | -2.714 | r = -0.34 | 0.116 | **0.007** |
| Intrapartum antibiotics [a] | -3.270 | r = 0.34 | 0.116 | **0.001** |
| Vaginal Streptococcus agalactiae positive [a] | -2.083 | r = 0.26 | 0.067 | **0.037** |
| Time of the second stage of labour [b] | -0.274 | r = -0.27 | 0.075 | **0.050** |
| Length of hospital stay [b] | 0.217 | r = 0.22 | 0.047 | **0.026** |

[a]Mann–Whitney-U,

[b]Rho-Spearman test.

Neonates with omphalitis presented mild umbilical redness (n = 4, 100%), umbilical oedema (n = 2, 50%), serous drainage (n = 2, 50%), purulent discharge (n = 1, 25%), local heat (n = 0, 0%), local pain (n = 2, 50%), umbilical bad smell (n = 4, 100%), and positive culture of microorganisms by drainage and blood (n = 1, 25%). These newborns had no other complications. Table 5 shows the results in terms of the evaluation criteria for omphalitis.

Other associated complications in the first month of life of newborns appeared. Considering that the slight bleeding was very light (never active bleeding) and that the appearance of umbilical hernia is not associated with the healing process of the umbilical cord, it can be considered that 91.4% did not present a complication in the healing of the umbilical cord (Table 5).

Analysing the appearance of complications in the healing of the cord according to the healthcare provided, we found that the **presence of mild umbilical redness was** associated with vaginal delivery (p = 0.01, Chi = 6.56, Phi = 0.25). Newborns born vaginally are four times more likely to have umbilical erythema than those born by Caesarean section (OR = 4.13, CI 1.32–14.06).

**Table 5. Evaluation criteria for omphalitis and complications after separation.**

| | N (%) |
|---|---|
| *Criteria for omphalitis* | |
| Mild umbilical redness | 46(43.4%) |
| Serous drainage | 4(3.8%) |
| Purulence at the umbilicus | 1(0.9%) |
| Umbilical edema | 2(1.9%) |
| Local heat | 0(0%) |
| Local pain | 2(1.9%) |
| Umbilical bad smell | 4(3.8%) |
| Severe umbilical redness | 4(3.8%) |
| Organisms cultured from drainage | 1(0.9%) |
| Organisms cultured from blood | 1(0.9%) |
| *Umbilical complications* | |
| No events | 45(42.9%) |
| Slight bleeding | 48(45.7%) |
| Non-bleeding umbilical granuloma | 3(2.9%) |
| Slightly bleeding umbilical granuloma | 6(5.7%) |
| Umbilical hernia | 3(2.9%) |

Statistically significant differences were found between the occurrence of slight umbilical bleeding after umbilical cord separation as a function of navel coverage with the diaper (p = 0.017, Chi = 5.7, Phi = 0.233, OR = 3.683, CI: 1.21–11.25).

The occurrence of omphalitis was related to skin-to-skin contact at birth (Fisher test, p value = 0.04, Chi = 8.006, Phi = 0.275). Newborns with skin-to-skin contact with the mother were almost 12 times more likely to have omphalitis than those who did not (OR = 11.750, CI: 1.46–94.87). The association between the presence of mild umbilical redness and omphalitis was statistically significantly different (p = 0.048, Chi = 4.57, Phi = 0.21). Neonates with redness had a 51.8% probability of developing omphalitis (OR = 1.08, CI: 1.002–1.18).

## Umbilical cord separation time prediction

A total of 10 independent variables were entered into the multiple linear regression model: how often the umbilical cord gets wet, birth weight, breastfeeding, natural birth, season of the year: spring, Streptococcus agalactiae positive, intrapartum antibiotics, length of hospital stay, time of the second stage of labour, and Apgar. In this study, diagnosis of the model was made by independence of residues (Durbin–Watson = 1.998). The residues were independent (residue normality: Kolmogorov–Smirnov residue: p value = 0.447), indicating normality. Linearity was seen with partial regression graphs, which showed linearity. Regarding homoscedasticity, the graph showed standardised residue, which did not have any standard. Non-collinearity: variance inflation factor (VIF) and tolerance: There was no VIF value greater than 1.05, suggesting that there was no problem of collinearity in this study. Extreme values were analysed with Cook's Distance (= 0.000–0.110). The linear regression model fulfilled the diagnostic criteria. A statistically significant linear regression model was obtained (p = 0.000), which explains 43.9% of the variance of the data (F [5, 98]: 15.361, R = 0.663, $R^2$ = 0.439) (Table 6).

The multiple linear regression equation was as follows: umbilical cord separation time = 2349.347 + 644.708 how often the umbilical cord gets wet + 0.914 birth weight + 1194.846 + season of the year: spring + 1546.959 intrapartum antibiotics + 1344.446 Apgar 7–8.

## Discussion

The findings have been discussed as per the objectives of the study.

## Umbilical cord separation time

On the basis of the results of our study, the umbilical cord separation time with dry care obtained is similar to recent findings of other researchers: Authors found a median of 7 days (6–7 days) [25], and other researchers also obtained comparable times with a mean of 7.12 days [26] or a median of 6 days [27]. In the same vein, cord separation time had a mean of 7.54 ±2.37 days [28], and other authors also averaged 7.54±1.56 days [29]. Previous and thorough research has reported a mean of 6.58±2.16 days [30], as well as the finding of 6.1 ± 2.1 days [31] and a mean of 6.6 days [32]. However, this result was in contrast with research conducted in Bangladesh [33,34], reported shorter umbilical cord separation times with mean results of 4.83±2.05 and 4.78±1.82, respectively. In Mullany et al. [34] research, the cause may be that the separation time was measured approximately.

Other researchers have obtained longer umbilical cord separation times, as well as the findings of a mean of 10 days [35] and a median of 11.7 days [15]; these results could be explained by washing the umbilical area twice a day in both studies. Quattrin et al. [36] also reported an umbilical cord separation time of 10±4 days. There is a possibility that these results are also due to the locations where these studies were conducted, as they have the same climate

**Table 6. Multiple linear regression analysis of umbilical cord separation time.**

| Variables | Unstandardised Coefficients | | Standardised Coefficients | t | p-value | Partial correlation |
|---|---|---|---|---|---|---|
| | B | Std. Error | Beta | | | |
| Constant | 2349.347 | 1877.788 | 1.251 | | .214 | |
| How often the umbilical cord gets wet | 644.708 | 96.376 | .514 | 6.690 | **.000** | 0.560 |
| Birth weight | .914 | .550 | .126 | 1.662 | .100 | .166 |
| Season of the year: Spring | 1194.846 | 545.088 | .167 | 2.192 | **.031** | .216 |
| Intrapartum antibiotics | 1546.959 | 529.262 | .227 | 2.923 | **.004** | .283 |
| APGAR 7–8 | 1344.446 | 669.979 | .155 | 2.007 | **.048** | .199 |

according to the Köppen–Geiger climate classification, which differs from the dry climate in our research area.

## Umbilical cord care and separation time

In accordance with our findings on how often the umbilical cord gets wet, results obtained by other authors show that the umbilical cord takes less time to separate when it is dryer [37]. However, others researchers have reported longer cord separation times when the umbilical cord got wet twice a day [15,35], or three times a day [38]. Daily bathing of the newborn also delayed the umbilical cord separation time in other studies [39,40].

## Socio-demographic and clinical variables of newborns and separation time

The findings of this study show that the umbilical cord separation time is shorter in vaginally delivered neonates; this was in accordance with the current literature [1,41,42]. Neonates delivered by Caesarean section tend to have a longer cord separation time due to less bacterial colonization after birth; fewer leukocytes are attached to the umbilical cord [43], and the cord separation is mediated by leukocytes [44,45]. Other studies have had conflicting results in this regard, showing that there was no statistical relationship between cord separation time and type of delivery [29,46]. Other researchers have obtained a significant difference in terms of the type of delivery and found that, using covariant analysis, the difference in umbilical cord separation time was not affected by this factor [30], similar to our results in multivariate analysis.

No studies have been found in the literature comparing umbilical cord separation time with the presence of Group B Streptococcus in the vagina. Our results indicate that it delays the separation time, which may be due to the adhesion of the microorganism in the umbilical vein endothelium [47] or the concomitant use of antibiotics in these cases. Multivariate analysis showed that this variable has no statistical significance. Regarding the time of the second stage of labour, our results showed a negative correlation of association, and we can only speculate that prolonged delivery could influence local processes at the cord or general immune response. This hypothesis was also raised by other authors in view of their results [42]. Regarding the length of hospital stay, our results do not coincide with other authors who conclude that the cord separation time is not associated with the hours of hospitalization [27]. It should be noted that the multivariate analysis showed that this variable leaves the regression model.

## Umbilical cord healing complications

Regarding the appearance of omphalitis in developed countries, our results are similar to those obtained by other authors, who showed incidences of omphalitis of 2.1% [42], 2% [48], and 1.5% [25]. Other authors have noted results of 1% omphalitis in countries with a very high or

high human development index [28,36,49,50]. A study in Spain reported an incidence of 0.8% of omphalitis, which has been increasing in recent years, according to the authors [51], similarly to the 0.9% incidence of moderate omphalitis shown in our results. In contrast, we discovered that researchers reported incidences of omphalitis of 0% [52] but the criteria for diagnosis of omphalitis were based on criteria of hospital readmission for omphalitis: need for systemic antibiotics related to the cord, pustules around the cord site, or 'scalded skin syndrome'; however, 3% needed topical antibiotics. Others researchers reported incidences of 0.08% [35] and 0.26% [53], whose researches had important sample sizes.

The presence of mild umbilical redness does not coincide with the results obtained by other studies, with values of 0.8% [52] or 7% [36]. Other authors have obtained a figure of 70% [54]. Our data are closer to the result of 30.5% [55]. None of these authors indicated the degree of redness.

## Umbilical complications after separation

In our research, we had no cases of bleeding while the umbilical cord was connected to the neonate's abdomen; other authors obtained similar results [30]. Other researchers reported cord bleeding of 1.98% [52] and 10.6% [49]. Our results show 45.7% slight bleeding (just a few drops) after umbilical cord separation.

The incidence of granuloma in our research is very similar to that found by researchers at 8.7% [56] and 11.7% [42]. Studies have yielded conflicting results, with lower results such as 1–2% [36,54]. No studies found have compared the presence of umbilical redness with the type of delivery or with covering the navel with the diaper. We have also found no studies comparing the association between skin-to-skin contact with omphalitis or the presence of mild redness with omphalitis.

## Umbilical cord separation time prediction

Literature supports the umbilical cord separation time predictors reported by our research:

How often the umbilical cord gets wet has already been considered in previous paragraphs. In terms of birth weight, similar results were obtained by other authors, who reported that cord separation time increased with increasing birth weight, but, as in our study, the effect size was low [27]. In contrast, in some studies, multiple regression analysis showed no significant effect of birth weight on the time to umbilical cord separation [57]. Other authors have showed that there was a negative correlation between umbilical cord detachment time and birth weight of the infants in the pre-term group, but this correlation was not significant in term infants [58]. Regarding the administration of intrapartum antibiotics, our results indicate that their use delays the cord separation time. It is known that these antibiotics given to the mother will continue to be present in the amniotic fluid and umbilical blood [59,60]. Maternal intrapartum antibiotic treatment continues to exert a bactericidal effect on the umbilical cord and peripheral venous blood of newborn infants [61]. This correlation has also been described by researchers who antibiotic use itself or the reason for antibiotic use might have an impact on the infiltration of the stump by polymorphonuclear leucocytes [45,58]. However, other authors have showed that in the neonatal ICU, in pre-term infants, ante-natal antibiotic and corticosteroid use did not affect cord detachment time [58]. Only one study has been found that considers umbilical cord separation time according to the season of the year, which concludes that in spring the cord takes less time to detach [40]. In the area where that study was conducted, spring weather conditions are similar to summer in our geographical area. Our results indicated that umbilical cord separation time increased in spring, while it is true that the mean cord separation time in our research is shorter in summer than in the rest of the seasons, but

these differences were not statistically significant. Neonates with Apgar scores of 7–8 one minute after birth have longer separation times than those with a higher score; these results coincide with other authors [42], who show that neonates with a 10-min Apgar score of 10 had a significantly shorter cord separation time than those with an Apgar score < 10.

## Conclusions and implications

The WHO in developed countries recommends dry cord care together with the use of soap and water when the cord is soiled, and our findings support these recommendations. Dry umbilical cord care is a safe practice that soon detaches the umbilical cord, taking into account the factors studied that will vary the length of time until the umbilical cord is separated. The profile of a newborn with the fastest cord separation time would be the one who gets wet the least, the one with an Apgar score > 8 at birth, the one whose mother does not receive intrapartum antibiotics, the one who is not born in spring, and the one who has a smaller weight at birth. Nevertheless, it is necessary to assess whether the increased incidence of omphalitis may be due to increased diagnostic sensitivity or a real problem in clinical practice.

The major practical application is family intervention that will consist of health education on the factors influencing umbilical cord detachment, especially wetting recurrence, where caregivers can intervene. Moreover, pointing out that the most frequent complication is umbilical granuloma would help to reduce the anxiety of families and professionals about the natural process of umbilical cord separation, in terms of healing characteristics and separation time. Increasing confidence in the practice of dry cord care contributes to the social learning of the unnecessary application of products for umbilical cord healing in developed countries.

Future lines of research would be to replicate the study in similar contexts in order to carry out a cross-validation of the multiple linear regression model and thus be able to make forecasts. At the same time, by increasing the sample, potentially unknown predictors can be highlighted that may affect umbilical cord separation time. Additionally, qualitative studies are recommended to explore the satisfaction and fears of the newborns' families with the dry cord care.

## Supporting information

**S1 File. Dataset.**
(XLS)

## Author Contributions

**Conceptualization:** María Dolores López-Medina, Isabel María López-Medina.

**Data curation:** María Dolores López-Medina, Isabel María López-Medina.

**Formal analysis:** María Dolores López-Medina, Ana Belén López-Araque.

**Investigation:** María Dolores López-Medina, Ana Belén López-Araque.

**Methodology:** María Dolores López-Medina, Ana Belén López-Araque.

**Project administration:** María Dolores López-Medina.

**Resources:** María Dolores López-Medina.

**Software:** María Dolores López-Medina.

**Supervision:** Ana Belén López-Araque, Manuel Linares-Abad, Isabel María López-Medina.

**Validation:** María Dolores López-Medina, Isabel María López-Medina.

**Visualization:** María Dolores López-Medina.

**Writing – original draft:** María Dolores López-Medina.

**Writing – review & editing:** María Dolores López-Medina, Ana Belén López-Araque, Manuel Linares-Abad, Isabel María López-Medina.

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
