## [Decision Letter · Decision Letter 0]

15 Oct 2019

PONE-D-19-16940

Umbilical cord separation time, predictors and healing complications in newborns with dry care

PLOS ONE

Dear Dr. María Dolores López Medina,

Thank you for submitting your manuscript to PLOS ONE. After careful consideration, we feel that it has merit but does not fully meet PLOS ONE’s publication criteria as it currently stands. Therefore, we invite you to submit a revised version of the manuscript that addresses the points raised during the review process.

We would appreciate receiving your revised manuscript by Nov 29 2019 11:59PM. To enhance the reproducibility of your results, we recommend that if applicable you deposit your laboratory protocols in protocols.io, where a protocol can be assigned its own identifier (DOI) such that it can be cited independently in the future. For instructions see: http://journals.plos.org/plosone/s/submission-guidelines#loc-laboratory-protocols

We look forward to receiving your revised manuscript.

Kind regards,

Anna Palatnik, M.D.

Academic Editor

PLOS ONE

Journal Requirements:

Additional Editor Comments (if provided):

Reviewers' comments:

Reviewer's Responses to Questions

**Comments to the Author**

1. Is the manuscript technically sound, and do the data support the conclusions?

Reviewer #1: Yes

Reviewer #2: Yes

2. Has the statistical analysis been performed appropriately and rigorously? 

Reviewer #1: Yes

Reviewer #2: I Don't Know

3. Have the authors made all data underlying the findings in their manuscript fully available?

Reviewer #1: Yes

Reviewer #2: Yes

4. Is the manuscript presented in an intelligible fashion and written in standard English?

Reviewer #1: Yes

Reviewer #2: No

5. Review Comments to the Author

Reviewer #1: This manuscript prospectively studies the separation time of umbilical cord stump and risk for infection with routine soap and water/dry cord care. The current standard of care and recommendation from major clinical centers (NIH and AAP) supports soap and water/dry cord care, but parents still use antiseptic treatments because of lack of confidence with these recommendations. The authors chose to study this issue prospectively and compare their rates of infection and other complications. This topic has been extensively investigated before by other groups but this group effectively describe their concerns about parental refusal to believe these recommendations. Studying this issue in their region could help the doctors in that region families to follow the stated guidelines.

The manuscript is written well, easy to read, the English is excellent. The authors represent the time of separation is different ways in the abstract vs manuscript which was a bit confusing (I would recommend using days rather than seconds).

Overall, on first read this didn't appear to add more to the literature given that the evidence for the stated guidelines is strong, but perhaps the study using regional factors will help guide care and improve understanding of the mistrust of families of their study population.

Reviewer #2: There are several areas that require revision for minor grammatical errors. For example, the World Health Organization is written as both the World Health Organization and the World Health Organisation. In addition, some of the phrasing is a little awkward, especially when using passive voice.

In the introduction, there is a nice argument to support the importance of this research due to the large public health concern of neonatal infection. In particular, omphalitis can have considerable morbidity and mortality, a risk that increases if the umbilical cord remains attached and theoretically is then reduced if the umbilical cord detaches more quickly. This is seen in line 66 and 67. Consider including the reference for this statement that connects the time of umbilical cord attachment and increased risk of omphalitis, especially if there is any statistical information where a particular length of time increases the risk of omphalitis by a known amount. This would help further support the overall clinical importance of this research because even though the time of cord separation was statistically different in some cases this might not result in a clinically significant length of time to the neonate. For example, though there is a clear statistical difference of about 2 days between the groups of neonates whose umbilical cords became wet 0-4 times vs. >5 times, does 2 days make a clinical significance to these children? However, if the aim was merely to show that the clean and dry method is both safe and financially sound then the current information provided would be sufficient.

Please discuss the training or role of the researcher who is taking the measurements for the umbilical cord and providing a clinical assessment/ possible diagnosis of omphalitis. Are they a physician? Nurse?

In line 151, please provide the definition that the paper uses for late separation of the umbilical cord.

On page 8, within the socio-demographic and clinical newborn variables, please list gestational age as a length of weeks and days. For example, 40 weeks and 0 days instead of 280 days.

Finally, please state clearly that the umbilical cord separation time of interest in your research is the time when the umbilical cord naturally separates from the infant. This will clear up any questions that the reader might have when reading the title or abstract and wondering if the authors meant the time of clamping the cord from the mother at birth (delayed cord clamping) or something even more extreme such as the lotus birth where the placenta is never clamped and remains attached to the infant for several days.

6. PLOS authors have the option to publish the peer review history of their article (what does this mean?). If published, this will include your full peer review and any attached files.

Reviewer #1: No

Reviewer #2: No

---

## [Author Response · Author response to Decision Letter 0]

23 Oct 2019

Reviewers' comments:

• Reviewer #1: 

-Comment 1: “The authors chose to study this issue prospectively and compare their rates of infection and other complications. This topic has been extensively investigated before by other groups but this group effectively describe their concerns about parental refusal to believe these recommendations. Studying this issue in their region could help the doctors in that region families to follow the stated guidelines”.

Response 1:

The study population is reluctant to change, most of the research available in the scientific literature is from other regions, and this study will help us provide evidence-based care and improve understanding of the mistrust of families in our study area. 

-Comment 2: “The authors represent the time of separation is different ways in the abstract vs manuscript which was a bit confusing (I would recommend using days rather than seconds)”.

Response 2:

Umbilical cord separation time changed in the manuscript and in the abstract using days, to make it understandable.

• Reviewer #2: 

-Comment 1: “The World Health Organization is written as both the World Health Organization and the World Health Organisation”.

Response 1: 

Corrected grammatical errors; the World Health Organization name changed and others minor grammatical errors.

-Comment 2: “some of the phrasing is a little awkward, especially when

using passive voice”.

Response 2: 

Modified part of the phrasing from passive voice to simple past in the manuscript:

 o Line 72: Omphalitis is an infection of the umbilical cord stump.

 o Line 83-85: Estimated that the incidence of omphalitis in newborns in developed countries is around 0.7% and rises 

 to 6–8% in developing countries.

 o Line 159-162: The different actions carried out in the dry umbilical cord care: how often the umbilical cord gets wet, 

 pH of soap used, type of bath for neonates, wearing a garment that tightens the newborn's body, placing the 

 newborn's diaper under the umbilical cord, and covering the umbilical cord with clean gauze.

 o Line 169: In table 1, we can see the independent secondary variables of the studied population.

 o Line 225-226: Jaén Research Ethics Committee (26 November 2015) approved the study.

 o Line 261: In table 3, we can see the correlation between umbilical cord care and separation time.

 o Line 407-408: other authors obtained similar results [30].

 o Line 417-418: Literature supports the umbilical cord separation time predictors reported by our research.

-Comment 3: “In particular, omphalitis can have considerable morbidity and mortality, a risk that increases if the umbilical cord remains attached and theoretically is then reduced if the umbilical cord detaches more quickly. This is seen in line 66 and 67. Consider including the reference for this statement that connects the time of umbilical cord attachment and increased risk of omphalitis, especially if there is any statistical information where a particular length of time increases the risk of omphalitis by a known amount. This would help further support the overall clinical importance of this research because even though the time of cord separation was statistically different in some cases this might not result in a clinically significant length of time to the neonate”.

Response 3:

We have added a reference to lines 66-68. A possible consequence of bacterial colonisation is cord stump infection, a factor that can greatly increase morbidity and mortality. The risk of infection increases until the stump detaches [4].

-Comment 4: “For example, though there is a clear statistical difference of about 2 days between the groups of neonates whose umbilical cords became wet 0-4 times vs. >5 times, does 2 days make a clinical significance to these children? However, if the aim was merely to show that the clean and dry method is both safe and financially sound then the current information provided would be sufficient”.

Response 4:

We provide this information to know the factors involved in the procedure of the fall of the umbilical cord. There are no statistical data in the scientific literature to support our results. 

-Comment 5: “Please discuss the training or role of the researcher who is taking the measurements for the umbilical cord and providing a clinical assessment/

possible diagnosis of omphalitis. Are they a physician? Nurse?”

Response 5: 

We specify in several parts of the article who does the clinical assessment (line 150,156,178).

-Comment 6: “In line 151, please provide the definition that the paper uses for late separation of the umbilical cord”.

Response 6:

In line 151 (actually 153-156) we add a bibliographic reference that provides the definition for late umbilical cord separation.

-Comment 7: “On page 8, within the socio-demographic and clinical newborn variables, please list gestational age as a length of weeks and days. For example, 40 weeks and 0 days instead of 280 days”.

Response 7:

On page 8 we change the duration of gestational age to express it in weeks and days.

-Comment 8: “State clearly that the umbilical cord separation time of interest in your research is the time when the umbilical cord naturally separates from the infant. This will clear up any questions that the reader might have when reading the title or abstract and wondering if the authors meant the time of clamping the cord from the mother at birth (delayed cord clamping) or something even more extreme such as the lotus birth where the placenta is never clamped and remains attached to the infant for several days”.

Response 8:

In Line 148, 149 we add the definition of umbilical cord separation time: Umbilical cord separation time. Is time when the umbilical cord naturally separates from the infant.

---

## [Editor Report · Decision Letter 1]

16 Dec 2019

Umbilical cord separation time, predictors and healing complications in newborns with dry care

PONE-D-19-16940R1

Dear Dr. López Medina,

We are pleased to inform you that your manuscript has been judged scientifically suitable for publication and will be formally accepted for publication once it complies with all outstanding technical requirements.

With kind regards,

Anna Palatnik, M.D.

Academic Editor

PLOS ONE
---

## [Editor Report · Acceptance letter]

26 Dec 2019

PONE-D-19-16940R1 

Umbilical cord separation time, predictors and healing complications in newborns with dry care 

Dear Dr. López-Medina:

I am pleased to inform you that your manuscript has been deemed suitable for publication in PLOS ONE. Congratulations! Your manuscript is now with our production department. 

With kind regards,

on behalf of

Dr. Anna Palatnik 

Academic Editor

PLOS ONE